# Diffusion Kinetics Theory of Removal of Assemblies' Surface Deposits with Flushing Oil

Michael Vigdorowitsch [1,2,*]  , Valery V. Ostrikov [2], Alexander N. Pchelintsev [3]   and Irina Yu. Pchelintseva [4]

1 Angara GmbH, Mörsenbroicher Weg 191, 40470 Düsseldorf, Germany
2 All-Russian Scientific Research Institute for the Use of Machinery and Oil Products in Agriculture, Novo-Rubezhnyy Sidestr. 28, 392022 Tambov, Russia; viitinlab8@bk.ru
3 Department of Higher Mathematics, Tambov State Technical University, Sovetskaya Str. 106, 392000 Tambov, Russia; pchelintsev.an@yandex.ru
4 Department of Automated Decision Support Systems, Tambov State Technical University, Sovetskaya Str. 106, 392000 Tambov, Russia; irina_yu_10@mail.ru
* Correspondence: dr.vigdorowitsch@angara-gmbh.de

**Abstract:** The diffusion kinetics theory of cleaning assemblies such as combustion engines with flushing oil has been introduced. Evolution of tar deposits on the engine surfaces and in the lube system has been described through the erosion dynamics. The time-dependent concentration pattern related to hydrodynamic (sub)layers around the tar deposit has been uncovered. Nonlinear equations explaining the experimentally observed dependences for scouring the contaminants off with the oil have been derived and indicate the power law in time. For reference purposes, a similar analysis based on formal chemical kinetics has been accomplished. Factors and scouring parameters for the favor of either mechanism have been discussed. Any preference for either diffusion or chemical kinetics should be based on a careful selection of washing agents in the flushing oil. Future directions of studies are proposed.

**Keywords:** combustion engine; lube system; diffusion kinetics; detergent additives; hydrodynamic layers





## 1. Introduction

Solving the problem of regular removal of oxidation products [1–15] out of running engine with flushing oil, cleaning the lubrication system from contaminants without or under the influence of chemical reagents, the use of additional cleaning tools such as expendable filters and built-in centrifuges has are aimed at increasing the lifetime of machines and reducing the costs of lubricants and, generally, maintenance.

*Related studies.* In particular, a flushing oil possibly containing detergent additives can be used to remove contaminants from the engine and its lube system (oil ducts) [16]. Cleaning effectiveness is monitored by changes, in particular, in the oil color and by a reduction in sludge in the crankcase [17]. Detergents bind tars dissolved in the oil, soften both the hard tar deposits on the engine walls and the sediment having been formed, and ensure their effective flushing out. The properties of the liquid (oil) are essential and depend, in particular, on the additives detergency.

How the sediment is being washed out while cleaning the engine has been analyzed in experimental works (Figure 1). The time dependences of sediment content in the flushing oil are typical monotonously increasing curves. It is important that cleaning mostly takes place in the first half of the whole cleaning cycle. The later the cleaning is carried out, the less the amount of sediment removed with the flushing oil. This result is quite obvious, because the more sediment in the engine and crankcase, the greater sediment the amount washed out.

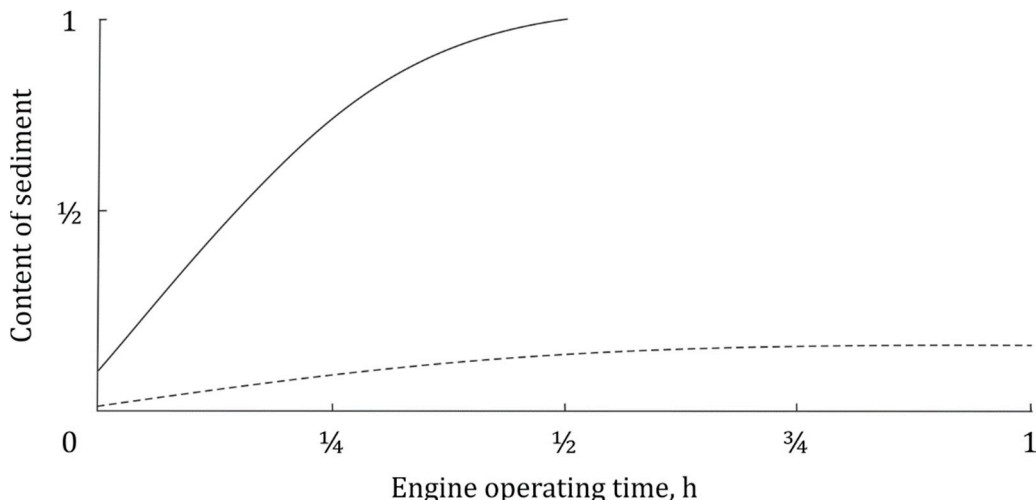

**Figure 1.** Change in the content of insoluble sediment in flushing oil. Diesel engine D-240. Flushing oil prepared on the basis of the waste oil recovered: engine oil M10G2 [18]; engine oil M10G2K after 200 h of running time, with 2 mass.% monoethanolamine, 2 mass.% isopropanol and 1–5 mass.% usual detergent additives added to recovered oil [17]. *See* source works for further details.

*Problem statement.* The authors have not found typical data with respect to removal of deposits from surfaces of engines or other assemblies with flushing oil in the literature, but the general character of relevant dependencies is certainly expected to resemble those in Figure 1. Of course, to effectively flush deposits such as tars, pyrobitumen and other types of varnish, the flushing oil should possess outstanding properties compared to those necessary to remove only the sediment. The cleaning with flushing oil should be thought of as related to either physical scouring of varnish or its chemical dissolution, or both. Some of the varnish components appear to be dissolved in the flushing oil and gone with it. The other components tend to form a sediment and will be caught by filters and possibly built-in centrifuge (typical for agricultural machines). Thereby, two phases of contaminants exist experiencing different hydrodynamic and other conditions during cleaning. In this paper, we will not consider fraction grading within the solid phase due to filters and centrifuges.

*The goal.* The goal of this paper is to propose a theoretical description of the diffusion and chemical kinetics of the cleaning process in running combustion engines, gears and other assemblies that does not seem to have been presented to the research community before. By addressing combustion engines hereinafter, we will imply a broader class of assemblies if applicable.

*Organization of the rest of this paper.* Section 2 presents the two models, one of which is the hydrodynamic scouring model of tar deposit removal (Section 2.1) and another is the chemical kinetics model (Section 2.2). A comparative discussion of results is given in Section 2.3. Section 3 is devoted to the tar removal equation in a macroscopic (statistical) formulation. Section 4 proposes conclusions and possible directions of future work.

## 2. The Models and Solutions

Let us consider how the flushing oil flow interacts with the combustion engine contaminants deposited on the surfaces of engine parts. The deposits have a complex composition, whose components, as a result of exposure to flushing oil, can be grouped in two categories: those (hereinafter subscript 1 assigned) forming the liquid phase in the oil and those (hereinafter subscript 2) floating in it as fine particles forming a slurry and contributing to the sediment. Thus, the components can be distinguished through their phases when in the flushing oil. For each component, there are two sequential processes: (*i*) detachment of the contaminant particles from the surface and (*ii*) transport with the flushing oil up to the discharge point (mechanical filtering devices for the solid phase). Within the deposit, the components are considered evenly distributed.

The deposit scouring turns out to be in many aspects similar to erosion of rocks by drilling fluids or groundwater, which has been repeatedly covered in the literature (see, e.g., [19,20]). It is important to distinguish the three scouring regimes—hydrodynamic scouring (often referred to as, e.g., diffusion kinetics regime or diffusion-controlled process), chemical dissolution (alternatively, kinetic or chemical kinetics regime) or a mixed process. We consider here in more detail the first two.

### 2.1. Hydrodynamic Scouring Model of Tar Deposit Removal

In consideration of the diffusion kinetics regime, we will employ the method [21] originally developed with respect to substance transfer along the flat solid surface with medium turbulence pulsations. We consider a solitary deposit as the hemisphere on the engine part surface, enabling us to qualitatively and accurately convey the hydrodynamics of the deposit erosion process without going into geometrical details (Figure 2). It is obvious that had we a deposit of a different shape under consideration, the resulting dependences would differ only by coefficients of the order of one. Unlike the diffusion kinetics model in [21], we have to treat sources of contaminants (deposits) somehow distributed on the flat solid surface. The deposits produce substance fluxes until exhausted, within the flushing oil. Taking into account a number of hydrodynamic features described in detail in [21], we will have to introduce some changes into the solving technique.

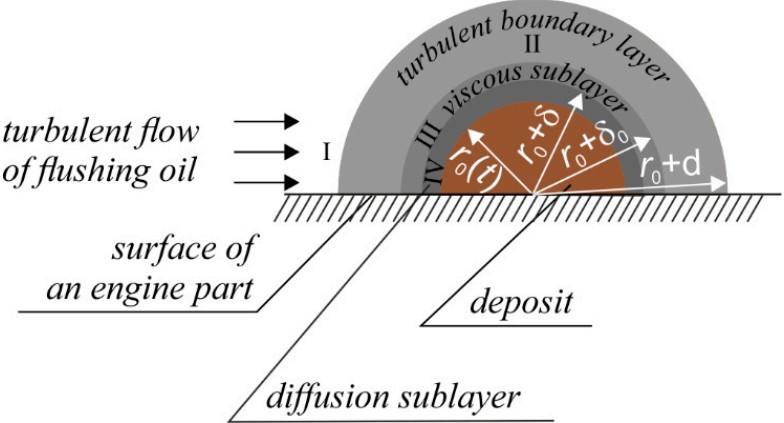

**Figure 2.** Hydrodynamic layers around the tar deposit being scoured.

Far from the deposit surface (I), there is a turbulent area where the main flow of flushing oil is attacking. All variables with subscript I will have hereinafter respective magnitudes in the bulk of flushing oil. Due to turbulence-related intense mixing, the concentration of the contaminant liquid phase in this area is independent of the distance to the deposit and equal to $C_{1,\text{I}}(t)$. The concentration of the solid phase there, on the other hand, is nearly zero because the phase is captured by the mechanical filter due to the circulation of the flushing oil: $C_{2,\text{I}}(t) \approx 0$. (The Dirichlet boundary condition is not ideal in this case. Indeed, it is about sinks somewhere at $r \to \infty$. However, a Neumann boundary condition would bring a practical difficulty because we could not then restrict our consideration by the deposits' neighborhood, the sink performance would depend on many factors including the centrifuge and filters properties, solid-phase fractions and the geometry of an engine and oil ducts, which would be difficult to estimate. With the Dirichlet boundary condition, we obtain somewhat smaller concentrations than in the outer layers and greater than in the inner sublayers but anyhow they all are tending to zero with the course of time (*see further*). The advantage is that we do not need to make arbitrary assumptions about the sink performance, etc.) In turbulent boundary layer II, diffusion and viscosity do not play a significant role. In viscous sublayer III, the turbulence becomes small and only contributes some corrections to the diffusion coefficient present in the Fick law. In the diffusion sublayer, the mass flux due to the deposit scouring obeys the Fick law with the true diffusion coefficient.

### 2.1.1. Distribution of Contaminant Concentrations

The hemispherical deposit with radius $r_0(t)$ is the source of substance flux into the bulk of flushing oil through the internal boundary of the diffusion sublayer. Being scoured, the deposit is decreasing in size, i.e., $0 < r_0(t_2) < r_0(t_1)$ if $t_1 < t_2$. We introduce coefficients $\nu_1$ and $\nu_2$ ($\nu_1 + \nu_2 = 1$), which mean that the mass of the deposit's $i$th component is equal to $m_i(t) = \nu_i m(t)$ ($i = 1, 2$), where $m$ is the mass of the entire deposit. Throughout this paper, subscript $i$ denotes the components within the deposit and their respective phases upon their transition into the flushing oil. Obviously, for the deposit mass: $m = m_1 + m_2$. On average, the ratio of masses of the contaminant liquid phase in the flushing oil and of the solid phase finally retained by the filters remains the same, depending on the deposit nature only.

The density $j_{0,i} \equiv j_i(r_0)$ of the deposit component flux the deposit produces at $r = r_0(t)$ is, by definition, equal to

$$j_{0,i} = -\frac{1}{s}\frac{dn_i}{dt} = -\frac{1}{2\pi r_0^2}\frac{dm_i}{M_i dt} \tag{1}$$

where $n_i = m_i/M_i$ is the amount of the $i$th component substance with molar mass $M_i$. Within both the diffusion and viscous sublayers, the deposit mass flux $J_i(r) = k\pi r^2 j_i(r)$ is invariant:

$$J_i(r) = J_i(r_0) = 2\pi r_0^2 j_i(r_0) \tag{2}$$

for $r_0 < r < r_0 + \delta_0$. Generally speaking, one could take into account retardation of the flux transfer between the sublayers, namely $J_i(r, t) = J_i\left(r_0\left(t - \frac{r-r_0}{\sqrt{D_i}}\right)\right)$ for $r_0 < r < r_0 + \delta$ and $J_i(r, t) = J_i\left(r_0\left(t - \frac{\delta}{\sqrt{D_i}} - \frac{r-r_0-\delta}{\sqrt{\widetilde{D}_i}}\right)\right)$ for $r_0 + \delta < r < r_0 + \delta_0$, where $D_i$ is the diffusion coefficient in the diffusion sublayer and $\widetilde{D}_i$ is that in the viscous sublayer. However, one is to give up this effect practically since the retardation is small because of effective scouring. Otherwise, ineffective scouring with inappropriate flushing oil would not make much sense, and one should change the flushing oil to increase scouring efficiency. Ignoring the retardation and according to Equation (2), we have the following equation for the deposit flux densities at different locations

$$j_i(r) \approx \frac{r_0^2}{r^2} j_{0,i} \tag{3}$$

In the diffusion sublayer, the Fick law is valid:

$$j_i(r) = -D_i \frac{\partial C_{i,IV}}{\partial r} \tag{4}$$

Combination of Equation (4) with Equation (3) and integration lead to

$$C_{i,IV}(r,t) = \frac{j_{0,i} r_0^2}{D_i}\frac{1}{r} + A_{i,1}$$

where integration constant $A_{i,1}$ is to be determined by means of boundary condition $C_{i,IV}(r_0, t) = C_{i,IV}^{(0)}$ representing the concentration of the respective phase in the deposit and to be determined by physicochemical analysis. Finally, for the concentration in the diffusion layer, we have

$$C_{i,IV}(r,t) = C_{i,IV}^{(0)} - \frac{j_{0,i} r_0}{D_i}\left(1 - \frac{r_0}{r}\right) \tag{5}$$

In the viscous sublayer, a different diffusion coefficient (one of possible models) $\widetilde{D}_i \sim \frac{v_0 r^4}{\delta_0^3}$ is used, where $v_0$ is a characteristic velocity of turbulent pulsations. The Fick law now reads

$$j_i(r) = -\gamma_i \frac{v_0 r^4}{\delta_0^3} \frac{\partial C_{i,\mathrm{III}}}{\partial r} \tag{6}$$

where $\gamma_i$ is a dimensionless factor nearly equal to 1. Substitution of Equation (3) into Equation (6) and integration result in the equation

$$C_{i,\mathrm{III}}(r,t) = \frac{j_{0,i} \delta_0^3 r_0^2}{5 \gamma_i v_0} \frac{1}{r^5} + A_{i,2}$$

Integration of constant $A_{i,2}$ is to be determined by the concentration continuity at $r = r_0 + \delta$:

$$C_{i,\mathrm{III}}(r,t) = C_{i,\mathrm{IV}}^{(0)} - \frac{j_{0,i}}{D_i} \frac{r_0 \delta}{r_0 + \delta} - \frac{j_{0,i} \delta_0^3 r_0^2}{5 \gamma_i v_0} \left[ \frac{1}{(r_0 + \delta)^5} - \frac{1}{r^5} \right] \tag{7}$$

In the turbulent boundary layer, the flux density appears to be independent of $r$ because of turbulence, the turbulent "diffusion coefficient" depends on the distance $r - r_0$ to the deposit center on the engine part surface, and the analogue of Equations (4) and (6) reads

$$j_i = -\beta_i v_0 (r - r_0) \frac{\partial C_{i,\mathrm{II}}}{\partial r}$$

where $\beta_i$ are some dimensionless factors nearly equal to 1. Integration brings us to the equation

$$C_{i,\mathrm{II}}(r,t) = \frac{j_i}{\beta_i v_0} \ln \frac{r - r_0}{d} + A_{i,3}$$

Integration of constant $A_{i,3}$ is used to construct a solution where it is associated with the bulk concentration $C_{i,\mathrm{I}}(t)$ of the substance:

$$C_{i,\mathrm{II}}(r,t) = \frac{j_i}{\beta_i v_0} \ln \frac{r - r_0}{d} + C_{i,\mathrm{I}}(t)$$

Flux density $j_i$ follows from the concentration continuity condition at $r = r_0 + \delta_0$:

$$j_i = \frac{-C_{i,\mathrm{IV}}^{(0)} + \frac{j_{0,i}}{D_i} \frac{r_0 \delta}{r_0 + \delta} + \frac{j_{0,i} \delta_0^3 r_0^2}{5 \gamma_i v_0} \left[ \frac{1}{(r_0 + \delta)^5} - \frac{1}{(r_0 + \delta_0)^5} \right] + C_{i,\mathrm{I}}}{\ln \frac{\delta_0}{d}} \beta_i v_0$$

For the concentration, then finally have:

$$C_{i,\mathrm{II}}(r,t) = \left\{ C_{i,\mathrm{IV}}^{(0)} - \frac{j_{0,i}}{D_i} \frac{r_0 \delta}{r_0 + \delta} - \frac{j_{0,i} \delta_0^3 r_0^2}{5 \gamma_i v_0} \left[ \frac{1}{(r_0 + \delta)^5} - \frac{1}{(r_0 + \delta_0)^5} \right] - C_{i,\mathrm{I}} \right\} \frac{\ln \frac{r - r_0}{d}}{\ln \frac{\delta_0}{d}} + C_{i,\mathrm{I}} \tag{8}$$

Equations (5), (7) and (8) describe the concentration distribution in the neighborhood of the tar deposit being scoured. It should be noted that they withstand the limit transition at $r_0 \to 0$: in Equation (5), the 2nd term is equal to zero whereas $C_{\mathrm{IV}}^{(0)}$ without the substance is zero, and in Equation (7), the 2nd and 3rd terms are nullified; in Equation (8), construction with logarithms tends to 1, the three first terms in the curly brackets go to zero, and $\pm C_{1,\mathrm{I}}$ cancel one another out whereas $\pm C_{2,\mathrm{I}} = 0$ as mentioned above.

### 2.1.2. Dynamics of the Deposit Scouring

In the model proposed above, the deposit mass and size are interrelated through substance-specific density $\rho$ as

$$dm_i = \rho_i 2 \pi r_0^2 dr_0$$

The specific densities arise because the component substances are considered to be evenly and continuously distributed within deposits. The component volume $V_{i,\text{true}}$ of the component substance is not mixed with anything else (specific densities appear to be additive: $m = m_1 + m_2 = (\rho_1 + \rho_2)V$, and the specific density of the entire deposit is the result of averaging the true densities by volume: $\rho = \frac{(\rho_{1,\text{true}} V_{1,\text{true}} + \rho_{2,\text{true}} V_{2,\text{true}})}{\left(\frac{2}{3}\pi r_0^3\right)} = \frac{m_1 + m_2}{V} = \frac{m}{V}$) and is smaller than the deposit volume. The specific density is related to substance normal (true) density through equation $\rho_i\left(\frac{2}{3}\pi r_0^3\right) = \rho_{i,\text{true}} V_{i,\text{true}}$. So, taking into account Equation (1), we obtain

$$j_{0,i} = -\frac{\rho_i}{M_i}\frac{dr_0}{dt}$$

This flux density at the deposit boundary does not depend on time under the conditions of even scouring until the deposit is completely scoured, and it is determined by external factors such as flushing oil properties and the attacking flow velocity. The dependence $r_0 = r_0(t)$ originates thereby from the scouring efficiency controlled through measurements of $C_{i,\text{I}}(t)$. Integration leads to the following equation

$$r_0(t) = r_0^{(0)} - \frac{M_i}{\rho_i}j_{0,i}t \tag{9}$$

where integration constant $r_0^{(0)}$ is the hemispherical deposit radius at the beginning of flushing $(t = 0)$, $0 < t < \frac{r_0^{(0)}\rho_i}{M_i j_{0,i}}$. At a given time, calculated for any of the components (where both components are being eroded simultaneously because of their even distribution within the entire deposit),

$$t_* = \frac{\rho_i r_0^{(0)}}{M_i j_{0,i}} \tag{10}$$

and the deposit is completely scoured.

With Equation (9), we easily calculate the component mass still remaining within the deposit at time $t < t_*$:

$$m_i(t) = \frac{2\pi\rho_i}{3}\left(r_0^{(0)} - \frac{M_i}{\rho_i}j_{0,i}t\right)^3 \tag{11}$$

Its derivative represents the deposit erosion rate:

$$\frac{dm_i(t)}{dt} = -M_i \cdot 2\pi r_0^{(0)2}\left(1 - \frac{M_i}{\rho_i}\frac{j_{0,i}}{r_0^{(0)}}t\right)^2 \cdot j_{0,i}$$

This expression is very much demonstrative as it explicitly incorporates the molar mass, current deposit surface square and the flux density (all addressed items are separated from one another by two multiplication signs "·"). The mass $\Delta m(t)$ of the substance removed with the flushing oil is equal to

$$\Delta m_i(t) = m_{0,i} - m_i(t) = \frac{2\pi\rho_i}{3}r_0^{(0)3}\left[1 - \left(1 - \frac{M_i}{\rho_i}\frac{j_{0,i}}{r_0^{(0)}}t\right)^3\right] \tag{12}$$

where $m_{0,i} = m_i(t = 0)$. Thus, $\Delta m_i(0) = 0$ and $\Delta m_i\left(t = \frac{r_0^{(0)}\rho_i}{M_i j_{0,i}}\right) = m_{0,i}$. The same relations are obviously valid for $m_0$ and $\Delta m$. The derivative of $\Delta m_i(t)$ reads

$$\frac{d\Delta m_i(t)}{dt} = -\frac{dm_i(t)}{dt} \geq 0$$

In particular, it follows that the cleaning rate at the beginning of combustion engine cleaning is equal to

$$\frac{d\Delta m_i(t=0)}{dt} = M_i \cdot 2\pi r_0^{(0)2} \cdot j_{0,i} \tag{13}$$

and is only zero at $t = t_*$. Thus, the deposit scouring rate is monotonously decreasing in time and turns out to be zero when the deposit vanishes completely. At the final point in Figure 3, the deposit substance appears to have been transferred into the bulk of oil so that the deposit itself does not exist anymore.

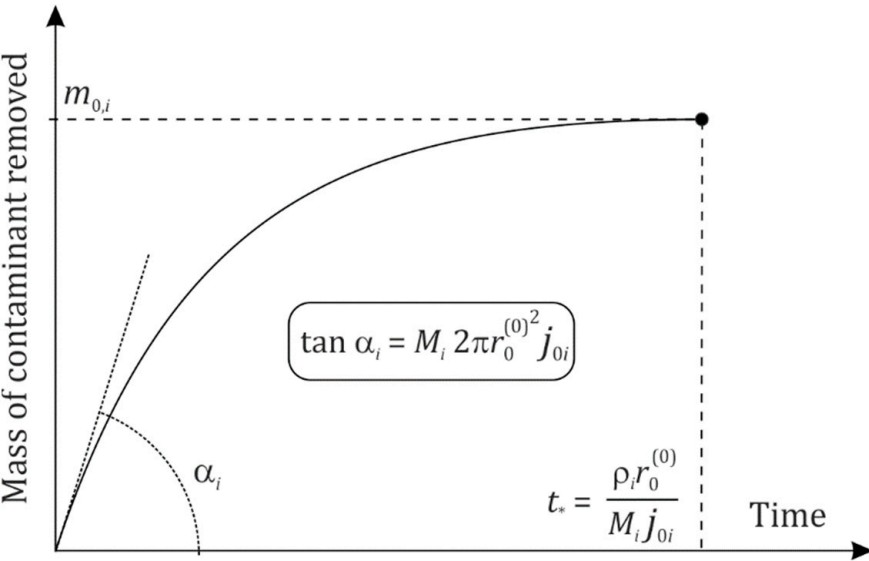

**Figure 3.** Dynamics of the removal of deposit contaminant components according to Equation (12).

One can calculate the mass distribution between the (sub)layers for the deposit scoured part (and that for its components). The expressions in quadrature for the diffusion, viscous and turbulent boundary (sub)layers read, respectively (omitting the component subscripts):

$$\Delta m_{\mathrm{IV}}(t) = 2\pi M \int_{r_0}^{r_0+\delta} C_{IV}(t) r^2 dr \tag{14}$$

$$\Delta m_{\mathrm{III}}(t) = 2\pi M \int_{r_0+\delta}^{r_0+\delta_0} C_{III}(t) r^2 dr \tag{15}$$

$$\Delta m_{\mathrm{II}}(t) = 2\pi M \int_{r_0+\delta_0}^{r_0+d} C_{II}(t) r^2 dr \tag{16}$$

The corresponding expression for the bulk of flushing oil follows from the mass conservation condition:

$$\Delta m_{\mathrm{I}}(t) = \Delta m(t) - \Delta m_{\mathrm{II}}(t) - \Delta m_{\mathrm{III}}(t) - \Delta m_{\mathrm{IV}}(t) \tag{17}$$

A bit cumbersome at first, the final expressions after substitution of Equations (5), (7) and (8) into Equations (14)–(17) are easy to obtain and omitted here because of their unclear practical value. With $r_0 \to 0$, $\Delta m_{\mathrm{II, III, IV}} \to 0$ and $\Delta m_{\mathrm{I}} \to \Delta m$ take place. Therefore, the capacity of the diffusion, viscous and turbulent boundary (sub)layers matters only at the beginning phase of cleaning the combustion engine.

### 2.2. Softening and Flushing of Tar Deposits in the Chemical Kinetics Model

The preliminary considerations for softening and scouring the solid tar deposits in the chemical kinetics model are as follows. When the reagent interacts with the deposit components chemically, both the formation of chemical reaction products and the release

of contaminants in the form of solids previously trapped during the deposit origination on the engine part surface to be cleaned occur in the turbulent medium. The products of the chemical reaction are expected to have a friable structure and be transported in dissolved or suspended form into the bulk of the flushing oil without diffusion limitations. Obviously, a different time function than that in Equation (12) is expected.

Potentially, we consider $l$ particles of the deposit substances, participating in the elementary act in a reaction with the flushing oil. Since the solutions for $l = 1$ and for integer $l \geq 2$ have different functional forms, we have to treat the respective cases separately in the framework of the formal chemical kinetics [22,23]. For the sake of completeness, selected cases are presented in Table 1, where $c_i$ is the reaction rate constant, and $C$ is the concentration of the reagent coming with the flushing oil (in the simplest case, we consider an excess of its reagent at a concentration that remains constant). A particular case, $l = 2$, is given as an example. In all cases, we begin with the corresponding kinetic equation, obtain its solution by employing the boundary condition $m_i(t = 0) = m_{0,i}$ and further calculate the mass of the deposit substance removed, as defined in Equation (12) and the deposit erosion rate.

**Table 1.** Kinetic equations, solutions and related functions for *l*-particle reactions (on the deposit side).

| | Structure of an Elementary Act from the Deposit Side. Cases: | | |
| --- | --- | --- | --- |
| | **1 Particle** | ***l* Particle ($l \geq 2$)** | **2 Particles (Example)** |
| Kinetic equation | $\frac{dm_i}{c_i dt} = -m_i C$ | $\frac{dm_i}{c_i dt} = -m_i^l C$ | $\frac{dm_i}{c_i dt} = -m_i^2 C$ |
| Solution $m(t)$ | $m_i(t) = m_{0,i} e^{-c_i C t}$ | $m_i(t) = \dfrac{m_{0,i}}{\left[1+(l-1)c_i C m_{0,i}^{l-1} t\right]^{\frac{1}{l-1}}}$ | $m_i(t) = \dfrac{m_{0,i}}{1+c_i C m_{0,i} t}$ |
| Deposit substance mass $\Delta m(t)$ removed | $\Delta m_i(t) = m_{0,i}\left(1 - e^{-c_i C t}\right)$ | $\Delta m_i(t) =$ $m_{0,i}\left\{1 - \dfrac{1}{\left[1+(l-1)c_i C m_{0,i}^{l-1} t\right]^{\frac{1}{l-1}}}\right\}$ | $\Delta m_i(t) =$ $m_{0,i} \times \left(1 - \dfrac{1}{1+c_i C m_{0,i} t}\right)$ |
| Deposit erosion rate $dm(t)/dt$ | $\frac{dm_i(t)}{dt} = -m_{0,i} c_i C e^{-c_i C t}$ | $\frac{dm_i(t)}{dt} = -\dfrac{c_i C m_{0,i}^l}{\left[1+(l-1)c_i C m_{0,i}^{l-1} t\right]^{\frac{l}{l-1}}}$ | $\frac{dm_i(t)}{dt} = -\dfrac{c_i C m_{0,i}^2}{(1+c_i C m_{0,i} t)^2}$ |
| $\Delta m(t)$ for small $t$ ♣ | $\Delta m_i(t) \approx$ $m_{0,i} c_i C t \times \left(1 - \frac{c_i C t}{2}\right)$ | $\Delta m_i(t) \approx m_{0,i}^l (l-1)^2 c_i C t \times$ $\left(1 - \frac{l(l-1)}{2} c_i C m_{0,i}^{l-1} t\right)$ | $\Delta m_i(t) \approx$ $m_{0,i}^2 c_i C t \times (1 - c_i C m_{0,i} t)$ |

♣ Small $t$: in the 1-particle case, $t \ll \frac{1}{c_i C}$; in the *l*-particle case, $t \ll \frac{m_{0,i}^{1-l}}{(l-1)c_i C}$; in the 2-particle case, $t \ll \frac{1}{c_i C m_{0,i}}$.

The dynamics of the contaminant mass is presented in Figure 4. With time, the deposit substance transferred into the bulk of the oil is increasing while the deposit itself is degrading. The derivatives of the dependencies have no zeros so that the curves asymptotically approach the horizontal line corresponding to the original mass of the deposit component. At $t = 0$, the growth of all the curves is described by the equation.

$$\frac{d\Delta m_i(t = 0)}{dt} = c_i C m_{0,i}^l \tag{18}$$

Its dissimilarity is represented by the power of $m_{0,i}$, which is the number of the deposit component particles participating in the elementary act. At big $t$, the curve in the case of a one-particle reaction tends toward the asymptotic line faster than that of $l \geq 2$ because the exponent decay overcomes tending $t^{-l+1}$ to zero with $t \to \infty$.

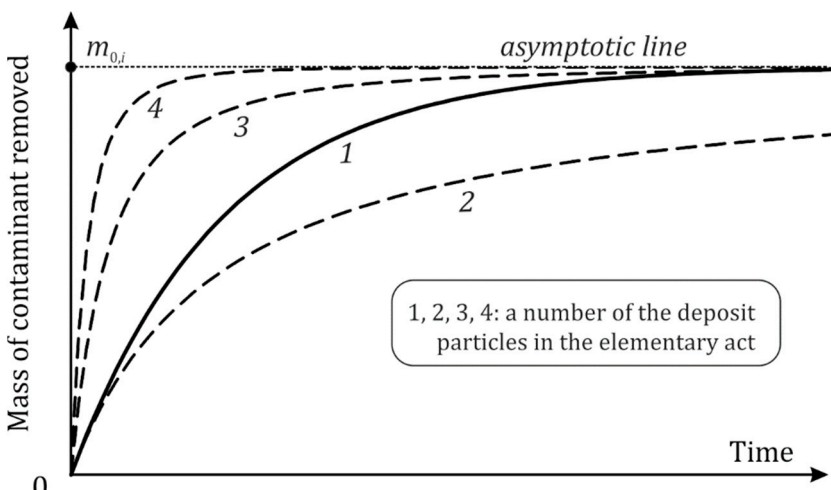

**Figure 4.** Dynamics of the removal of deposit contaminant components $\Delta m_i(t)$ according to Table 1. Solid line: exponential decay at big $t$; dashed lines: decrease according to a power law.

*2.3. Discussion*

The results presented above substantiate the experimental findings pointed out in the Introduction, and specify the behavior of curves, depending on the regime in which the flushing oil flow produces an impact on contaminant deposits. One has to distinguish between the diffusion kinetics, chemical reaction kinetics and a mixed regime.

In practical experiments and at service, the contaminant amount is typically determined on the basis of (*i*) the physicochemical analysis of the flushing oil and (*ii*) the mass and structure of the sediment caught by the built-in centrifuge and filters. The time dependence of the contaminant mass carries a message whether the cleaning is nearly to be completed. Characteristic in this sense are the curves presented in Figures 1, 3 and 4. Qualitatively, the theoretical regularities resemble the experimental curves in Figure 1 pretty well. Visually, it may appear hard to distinguish which of the two mechanisms takes place or dominates. In the case of diffusion kinetics (Section 2.1), removal of contaminants is proceeding according to the 3rd power of time with dimensional factors such as the molar mass of the deposit substance and its flux $j_0$ whose magnitude partly originates from the flushing oil properties, and other relevant conditions (Equation (11)) such as the turbulence intensity, and (self-)adhesion of the substance. With the formal chemical kinetics, one arrives at different laws of the deposit dissolution, which are the exponential decay in the case of a one-particle reaction (on the deposit side) or power function $t^{-\alpha}$ ($1 < \alpha \leq 2$), otherwise see Table 1. In the case of a one-particle chemical reaction, it is possible to identify the process mechanism since, in the semilogarithmic coordinates, the dependence $1 - \Delta m_i(t)/m_{0,i} = e^{-c_i C t}$ transforms into a straight line. Another difference between the regimes is related to the time to complete removal of the deposit, i.e., the diffusion kinetics regime defined as finite (Equation (10)), whereas in the case of a purely chemical dissolution, it appears to be mathematically infinite. As for the beginning of cleaning the engine in these two regimes, we refer to Equations (13) and (18):

$$\left.\frac{d\Delta m_i(t=0)}{dt}\right|_{\text{diff.k.}} \bigg/ \left.\frac{d\Delta m_i(t=0)}{dt}\right|_{\text{chem.k.}} = \frac{3 M_i j_{0,i}}{\rho_i} \frac{1}{c_i C} r_0^{(0)2-3l} \tag{19}$$

Should the purely chemical regime proceed a one-particle reaction, this expression takes the form $\frac{3 M_i j_{0,i}}{\rho_i} \frac{1}{c_i C} \frac{1}{r_0^{(0)}}$. The first factor refers to the deposit substance properties, the second to the reaction characteristics and the third is the reverse initial size of the deposit. A large molar mass, powerful substance flux, small density and small initial size make the diffusion kinetics more effective compared to the chemical kinetics whereas a high reaction

constant, a high concentration of the reagent in the bulk of flushing oil as well as large deposits promote the chemical kinetics more effectively compared to diffusion kinetics.

It is useful to keep in mind that as we consider a two-phase composition of the deposits, the following sum addresses measuring the mass of components transferred to the bulk of the oil and further up to the discharge point: $\Delta m = \Delta m_1 + \Delta m_2$, where the subscripts refer to the respective phases.

The final remark here is related to the circumstance where, although the finite time interval of deposit removal is always good, there are inappropriate turbulence conditions and other accompanying factors—this interval may appear rather big. Therefore, some preferences for diffusion kinetics (possibly, longer cleaning) or chemical kinetics (possibly, chemical impact on gaskets, etc.) may be a result of a delicate balance including careful selection of washing agents in the flushing oil.

### 3. Macroscopic Cleaning of Combustion Engine Part Surfaces

Equation (12) and the equations in Table 1 characterize a solitary tar deposit on the engine part surface. As such deposits vary in size and, consequently, in mass, the true macroscopic result of engine cleaning with flushing oil in terms of entire removed mass $\Delta M$ of all deposits to the time $t$ is

$$\Delta M(t) = \sum_{l=1}^{L} \Delta m \left( m_0^{(l)}, t \right) \delta n_l \left( m_0^{(l)} \right) \qquad (20)$$

where subscript $l$ enumerates deposits differing in their initial mass $m_0^{(l)}$ and runs from 1 up to some macroscopically big value $L$, $\delta n_l$ is the number of deposits of the corresponding mass, $\Delta m \left( m_0^{(l)}, t \right)$ is the $l$th deposit mass transferred to the bulk of the oil to the time $t$. The summation in Equation (20) is worth replacing with integration employing the technique introduced in [24,25]. The number of deposits $\delta n_l \left( m_0^{(l)} \right)$ cannot, for physical reasons, be a function of a discontinuity type and obviously belongs to a class of at least continuously differentiable functions. Moreover, if we assume that the summation over $l$ is ordered by the mass of the deposit, i.e., smaller $l$s correspond to deposits with smaller mass, then, up from a certain point as $l$ increases, the function $\delta n_l \left( m_0^{(l)} \right)$ will tend to zero because there are no deposits of infinitely large mass. Such a tendency towards zero is to be expected by an exponential law due to the normal (Gaussian) distribution. These considerations make the following transition from the sum to an integral reasonable:

$$\Delta M(t) \approx \int_0^\infty \Delta m(m_0, t) n(m_0) dm_0 \qquad (21)$$

Here, expecting an exponential decay of $n(m_0)$ while in the region of sufficiently big $m_0$ and with further growth of $m_0$, we have used an infinite upper limit instead of some $(m_{0,max} + \delta m)$, where an a priori unknown $\delta m > 0$. Equation (21) involves some mass distribution function $n = n(m_0)$, its dimension is $[m]^{-1}$.

The limit of Equation (21) at $t \to \infty$ mathematically means that the entire initial mass $M_0$ of all deposits accumulated on the engine part surfaces has passed into the bulk of the flushing oil:

$$\Delta M(t \to \infty) \to M_0$$

In the case of the diffusion kinetics model, it is sufficient to reach some $t_*$ corresponding to scouring of the largest deposits. Thus, at a sufficiently large $t$, function $\Delta M(t)$ reaches its greatest value or experiences asymptotic behavior. Based on $\Delta M(t)$ obtained from experiments, one is to judge how far it is from complete removal of deposits from the engine parts surface.

Thus, Equation (21) together with both Equation (12) and Table 1 gives a solution of the macroscopic problem of the removal of tar deposits out of the combustion engine. It should be noted, however, that tar deposits are complex compounds with respect to which

partial processes can be considered component wise. Therefore, the final result in the form of Equation (21) can be regarded as some averaging over a variety of deposits without specifying the features of complex compounds.

## 4. Conclusions

This paper introduces the theory of cleaning a combustion engine including its lube system, and possibly other relevant assemblies, with flushing oil. The resulting purity of the engine surfaces depends on many factors among which are cleaning time, turbulence degree, and the properties of both tar deposit substance and flushing oil. As a theoretical description of processes seems to have been absent, we undertook an attempt to propose a detailed hydrodynamical consideration in the framework of diffusion kinetics, followed by the formal chemical kinetics model, to compare the results of both approaches. As a qualitative reference point, we have referred to the experimental results [17,18] we depicted in Figure 1.

The nonlinear diffusion kinetics model enabled us to obtain a number of new results among which is the tar deposit erosion law (Equations (9) and (11)). The erosion rate incorporates the molar mass of the deposit substance, the current deposit surface square and the flux density depending on the deposit substance properties, turbulence degree and flushing oil properties. The mass of the contaminant substances being removed in the form of both liquid and solid phases is presented by Equation (12) and in Figure 3. The corresponding curve is monotonously increasing and finite in time. The time to complete removal of the contaminant from the engine surfaces is introduced by Equation (10). The time-dependent distribution of the deposit contaminants in the hydrodynamic layers is determined in terms of concentrations in Equations (5), (7) and (8) and in terms of masses in quadrature in Equations (14)–(17).

Engine cleaning is also considered from the stand-point of formal chemical kinetics. The same characteristics (when applicable) of the process are calculated in Table 1. The mass of contaminants transferred to the flushing oil and possibly (for the solid phase) to the filter or built-in centrifuge is presented in Figure 4. The time to complete transfer is infinite. However, this does not say much about the comparable effectiveness of cleaning in the diffusion and chemical kinetics regimes. Which one is preferable is a balance of a number of factors in Equation (19) related not only to removal of contaminants but also the integrity of gaskets.

Equation (21) was obtained, presenting in quadrature the current overall mass of contaminants transferred to the bulk of the oil, given a distribution of deposits on initial mass or size.

Future directions of work may include (*i*) a careful modeling of the chemical dissolution of tar deposits with flushing oil carrying surface-active additives, which would require a deeper adsorption-related analysis [23,26,27], compared to that in Table 1 and Figure 4, and (*ii*) a detailed investigation (with statistical methods) of the tar distribution on the engine part surfaces for credible calculations of the cleaning time required. In applications, the methodology developed in the present paper can find its place in service guides in the form of specifications of cleaning procedures.

**Author Contributions:** Conceptualization, V.V.O.; Methodology, M.V.; Validation, V.V.O.; Formal analysis, M.V., A.N.P. and I.Y.P.; Investigation, M.V.; Resources A.N.P.; Writing—original draft, M.V.; Writing—review & editing, M.V.; Supervision, V.V.O.; Project administration, V.V.O. All authors have read and agreed to the published version of the manuscript.

**Funding:** This research received no external funding.

**Data Availability Statement:** Data sharing not applicable.

**Conflicts of Interest:** The authors declare no conflict of interest.

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
