# Peer review of "Diffusion Kinetics Theory of Removal of Assemblies’ Surface Deposits with Flushing Oil"

_computation, doi:10.3390/computation11080164_

Round 1

Reviewer 1 Report

The diffusion kinetics theory of cleaning assemblies like combustion engines with flushing oil has been introduced. Evolution of tar deposits on the engine surfaces and in the lube system has been described through the erosion dynamics. The time-dependent concentration paÄ´ern related to  hydrodynamic [sub-]layers around the tar deposit has been uncovered. Nonlinear equations explaining the experimentally observed dependences for scouring the contaminants off with the oil have been derived. For the reference purposes, a similar analysis based on formal chemical kinetics  has been accomplished. Factors and scouring parameters for the favour of either mechanism have  been discussed and future directions of studies proposed. Generally, this is a good work. It can be accepted if the authors can consider the following issues: 1. There are a lot of equations in the paper. If the equations are not original, please give the references. 2. What is the motivation of the work? Why is the nonlinear model necessary? 3. The quality of figures should be improved.

The diffusion kinetics theory of cleaning assemblies like combustion engines with flushing oil has been introduced. Evolution of tar deposits on the engine surfaces and in the lube system has been described through the erosion dynamics. The time-dependent concentration paÄ´ern related to  hydrodynamic [sub-]layers around the tar deposit has been uncovered. Nonlinear equations explaining the experimentally observed dependences for scouring the contaminants off with the oil have been derived. For the reference purposes, a similar analysis based on formal chemical kinetics  has been accomplished. Factors and scouring parameters for the favour of either mechanism have  been discussed and future directions of studies proposed. Generally, this is a good work. It can be accepted if the authors can consider the following issues: 1. There are a lot of equations in the paper. If the equations are not original, please give the references. 2. What is the motivation of the work? Why is the nonlinear model necessary? 3.More related works are welcome to enrich the literature review such as A Survey of Intelligent Driving Vehicle Trajectory Tracking Based on Vehicle Dynamics;Event-Triggered Deep Learning Control of Quadrotors for Trajectory Tracking. 4. The quality of figures should be improved.

Author Response

In the file attached

Reviewer 2 Report

The process of cleaning combustion engines with flushing oil is studied. Diffusion kinetics theory of deposit removal is constructed. The model is consistent with experimental data.

In my opinion, the constructed theory is interesting and useful for both theorists and practical engineers. The formal disadvantage of the work is excessive self-citation.

Author Response

In the file attached

Reviewer 3 Report

This is an interesting article and can be considered for possible publication in computation journal after detailed major revision.

1. The title "Nonlinear dynamic models of removal of assemblies’ surface deposits with flushing oil" does not represent the novelty and contribution, please revised if possible

2. Abstract is two brief, please add the qualitative and quantitative infererces for better understanding of the contribution of the study

3. Introduction should be segmented in to related study section, problem statement with justification, contributions and organization of the rest of the paper. Please add few more reference for better justification of the problem statement (only 8 refs in the manuscript).

4. Methodology section with graphical abstract should be added.

5. All the graphical illustrations needs further interpretation of the results for better understanding of the readers. 

6. All graphical illustration are presented with abnormal font inside, it is suggested to reduce the font size inside all of the figures.

7. Format of the equations is not consistant, it should be presented on consistant pattern.

8. Whole manuscirpt is check carefully to present the information on consistant template for better understanding of the readers.

9. Future applications of the proposed methodologies should be listed in the conclusion section.

Author Response

In the file attached

Round 2

Reviewer 3 Report

No further comments